# Impact of *TP53* mutations in acute myeloid leukemia patients treated with azacitidine

**Pierre Bories**[1,2,3]*, **Naïs Prade**[1], **Stéphanie Lagarde**[1], **Bastien Cabarrou**[4], **Laetitia Largeaud**[1], **Julien Plenecassagnes**[5], **Isabelle Luquet**[1], **Véronique De Mas**[1], **Thomas Filleron**[4], **Manon Cassou**[5], **Audrey Sarry**[2], **Luc-Matthieu Fornecker**[6], **Célestine Simand**[6], **Sarah Bertoli**[2], **Christian Recher**[2], **Eric Delabesse**[1]

**1** Laboratoire d'Hématologie, Centre Hospitalier Universitaire de Toulouse, Institut Universitaire du Cancer de Toulouse Oncopole, Toulouse, France, **2** Service d'Hématologie, Centre Hospitalier Universitaire de Toulouse, Institut Universitaire du Cancer de Toulouse Oncopole, Toulouse, France, **3** Réseau Onco-occitanie, Institut Universitaire du Cancer de Toulouse Oncopole, Toulouse, France, **4** Unité de biostatistique, Institut Claudius Régaud, Institut Universitaire du Cancer de Toulouse Oncopole, Toulouse, France, **5** Unité de bioinformatique, Institut Claudius Régaud, Institut Universitaire du Cancer de Toulouse Oncopole, Toulouse, France, **6** Service d'Onco-Hématologie, Centre Hospitalier Universitaire de Strasbourg, Strasbourg, France

* pierre.bories@onco-occitanie.fr

**Data Availability Statement:** All relevant data are within the paper its Supporting Information files and data used for the analyses were deposited in

## Abstract

Hypomethylating agents are a classical frontline low-intensity therapy for older patients with acute myeloid leukemia. Recently, *TP53* gene mutations have been described as a potential predictive biomarker of better outcome in patients treated with a ten-day decitabine regimen., However, functional characteristics of TP53 mutant are heterogeneous, as reflected in multiple functional TP53 classifications and their impact in patients treated with azacitidine is less clear. We analyzed the therapeutic course and outcome of 279 patients treated with azacitidine between 2007 and 2016, prospectively enrolled in our regional healthcare network. By screening 224 of them, we detected *TP53* mutations in 55 patients (24.6%), including 53 patients (96.4%) harboring high-risk cytogenetics. The identification of any *TP53* mutation was associated with worse overall survival but not with response to azacitidine in the whole cohort and in the subgroup of patients with adverse karyotype. Stratification of patients according to three recent validated functional classifications did not allow the identification of *TP53* mutated patients who could benefit from azacitidine. Systematic TP53 mutant classification will deserve further exploration in the setting of patients treated with conventional therapy and in the emerging field of therapies targeting TP53 pathway.

## Introduction

With little improvement in their overall survival (OS) over the last decade, older patients with acute myeloid leukemia (AML) still harbor dismal prognosis [1, 2]. Validated therapeutic options are currently limited. Patient selection for intensive versus low-intensity therapy remains controversial [3, 4] and inter-physician practice variations are frequent, which underscores the uncertainty on the optimal strategy for any elderly AML patient [5].

Figshare, https://doi.org/10.6084/m9.figshare.12897077.v1.

**Funding:** This work was supported by the French government under the "Investissement d'avenir" program (ANR-11-PHUC-001). During the period of the study Stéphanie Lagarde has received salary from the "Investissement d'avenir" program (ANR-11-PHUC-001). Eric Delabesse's laboratory has received fundings from association 111 des Arts, association Laurette Fugain and Ligue Régionale contre le Cancer. Pierre Bories has received funding from the association L'Alsace Contre le Cancer for his PhD thesis. All these funders had no role in study design, data collection and analysis, decision to publish, or preparation of the manuscript.

**Competing interests:** Christian Recher has received research funding from Celgene, unrelated to this study. There are no patents, products in development or marketed products associated with this research to declare. This does not alter our adherence to PLOS ONE policies on sharing data and materials.

Although azacitidine failed to demonstrate its superiority in the AZA-AML-001 trial compared to intensive chemotherapy (IC) for patients older than 65 years with non-proliferative AML [6], several studies have found that patients with adverse cytogenetic risk myelodysplastic syndrome (MDS) or AML treated with hypomethylating agents (HMA) may obtain similar or even higher response rates than patients with intermediate-risk cytogenetics [7–9]. The prevalence of *TP53* mutations is typically extremely high (up to 50–70% in AML with complex karyotype [10–12]) in this population and the efficacy of HMA may reflect a TP53-independent mechanism of action. Alteration of TP53 functions is a well-known negative prognostic factor for MDS and AML patients treated with conventional chemotherapy [10, 13] and allogeneic stem cell transplantation [14–16], which has justified investigation of alternative TP53-independent therapy such as HMA. In preclinical studies, primary fibroblastic and for *TP53*-deficient neoplastic cells exhibit hypersensitivity to decitabine treatment compared to wild type cells, through apoptotic response [17, 18], illustrating a previously described concept of sensitization to apoptosis by the absence of *TP53* [19, 20] and extending this concept to HMA. A single-institution trial has described a protective effect of *TP53* mutations in AML patients treated frontline with an intensified regimen of decitabine [21]. However, only 21 patients (18%) harboring a *TP53* mutation were included in this trial, rendering necessary a confirmation of these interesting results. More recently, the pivotal AZA-AML-001 phase 3 trial described no significant association between *TP53* mutations and outcome of AML patients treated with azacitidine while *TP53* mutations remained associated with shorter OS in the conventional care comparator arm [22].

Although mutations in *TP53* have traditionally been considered functionally equivalent leading to a lack of function due the loss of the DNA-binding domain or mediated by a dominant-negative effect on the remaining functional wild-type allele, recently, some TP53 mutants were shown to display a gain of function (GOF) independent of wild-type *TP53* (TP53$^{wt}$) function [23]. Several classifications of *TP53* mutants have been proposed with a correlation to the patient outcome in solid tumors and diffuse large B cell lymphoma [24–26]. Their clinical usefulness has never been questioned in AML patients, although it might be highly important in the context of novel therapy targeting *TP53* and/or *MDM2*.

We took advantage of our extensively analyzed prospective regional AML registry [1, 27, 28] to investigate the impact of *TP53* mutations in a very large cohort of AML patients treated frontline with azacitidine. We further assessed the usefulness of *TP53* mutation classifications as a biomarker with the aim to eventually identify a sub-group of patients with *TP53* mutations who might specifically benefit from azacitidine.

## Patients and methods

### Regional cancer network ONCOMIP registry

AML patients (excluding M3) treated frontline with azacitidine were enrolled in the regional cancer network ONCOMIP registry between 2007 and 2016 [1, 27, 28] (Toulouse AML database). Patient's bone marrow samples were obtained following standard ethical procedures (Helsinki principles), after informed written consent, and stored at the HIMIP collection. According to the French law, the HIMIP collection was declared to the Ministry of Higher Education and Research (DC 2008–307 collection 1). The French Commission Nationale de l'Informatique et des Libertés (CNIL) authorised the use of patient data analyzed in our study Cytogenetic risk was assessed according to the MRC classification [29].

### *TP53* next generation sequencing

Genomic DNA (gDNA) was extracted from baseline bone marrow sample using a Qiagen DNA extraction kit (Qiagen). *TP53* status was derived from exome sequencing for 49 patients or using a Next Generation Sequencing multiplex PCR for 179 patients.

Exome capture was performed using Sureselect All-Exome V4 kit (Agilent). Exome libraries were then sequenced using a NextSeq500 sequencer (Illumina) and a SureSelect QXT Reagent kit (paired-end, 150bp, sequencing 2 x 150 cycles).

*TP53* Next Generation Sequencing was performed using a multiplex *TP53* PCR covering the complete coding sequences of exons 4 to 10 (primers listed in S1 Table). *TP53* libraries were then sequenced using a Miseq Reagent kit V2 (paired-end, 150bp, sequencing 2 x 150 cycles) and MiSeq sequencer (Illumina).

Alignment and variant calling were performed using NextGene software (SoftGenetics). *TP53* variants with a variant allele frequency (VAF) higher than 1%, were filtered using the *TP53* International Agency for Research on Cancer R18 database released in April 2016 (IARC) (http://www-p53.iarc.fr/) [30].

### TP53 mutation classifications

Each *TP53* mutation was analyzed according to 3 different classifications. For patients with more than one *TP53* mutation, we selected the mutation with the highest predicted impact [31].

**(1) Disruption classification [24].** This classification was based on the consequences of the *TP53* mutation on its protein folding, segregating disruptive versus non-disruptive mutations. This clustering was validated as a prognosis factor for OS in a series of head and neck carcinoma patients by Poeta *et al* [24]. It relies on the location of the mutation and the predicted amino acid alterations. Disruptive mutations are composed of (i) stop codons in any region or (ii) non-conservative mutations (i.e. change of category of the mutated amino-acid [non-polar as F, M, W, I, V, L, A and P; polar non charged as C, N, Q, T, Y, S and G; polar negatively charged as D and E and polar positively charged as H, K and R]) inside the key DNA-binding domains (L2–L3 region, corresponding to codons 163 to 195 and 236 to 251). All other mutations are classified as non-disruptive.

**(2) Evolutionary action TP53 Score classification.** Computational approach with the calculation of an evolutionary action score summarizing the phenotype to genotype relationship for each *TP53* missense mutation reliably stratified patients with head and neck cancers or metastatic colon cancers harbouring *TP53* mutations as high or low risk [25, 32]. The score was calculated for each *TP53* missense mutation identified in our cohort [25]. A threshold of 75 derived from the work of Neskey *et al.* was used to separate high risk from low risk *TP53* mutations.

**(3) Relative fitness score (RFS) classification.** Kotler *et al.* used a massively parallel proliferation assay deciphering the sequence to structure and function relationship of *TP53* mutations in human cells, leading to the establishment of a comprehensive catalogue of 9,833 unique *TP53* DNA-binding domain variants (DBD, corresponding to amino-acids 102 to 292) with their functions evaluated in human cells *in vitro* and *in vivo* [33] giving a relative fitness score for each of these TP53 mutant.

### Statistical analyses

The analysis date for the clinical evaluation of the database was June 1, 2018. All the data used for the analyses were deposited in Figshare: https://doi.org/10.6084/m9.figshare.12897077.v1. Clinical response was assessed using ELN criteria [34] after 3 and 6 cycles of azacitidine as

indicated. For patient who failed to achieve at least partial clinical remission, we also assessed hematological improvement using the MDS IWG 2006 response criteria [35].

Data were summarized using descriptive statistics. Categorical variables were presented as frequency, percentage and number of missing data. Continuous variables were presented as median, range and number of missing data. Comparisons between groups were performed using the Chi-squared or Fisher's exact test for categorical variables and the Mann-Whitney test for continuous variables. Duration of response was evaluated in patients achieving a response as defined by complete remission (CR) and complete remission with incomplete hematologic recovery (CRi). It was defined as the time from the date of response to the date of relapse or death from any cause and was estimated using the Kaplan-Meier method. Overall survival (OS) was defined as the time from the date of diagnosis to the date of death from any cause, patients alive were censored at last follow up news. Survival rates were estimated using the Kaplan-Meier method. Univariable and multivariable analyses were performed using the Logrank test and the Cox proportional hazards model; hazard ratios were estimated with their 95% confidence intervals. All tests were two-sided and p-values < 0.05 were considered statistically significant. Statistical analyses were conducted using STATA 13 (StataCorp, Texas, USA) software.

## Results

### Characteristics of patients treated with azacitidine

From January $1^{st}$ 2007 to December $31^{st}$ 2016, 279 AML treated frontline with azacitidine were enrolled in the regional cancer network ONCOMIP registry. Patients received a median of 6 cycles of azacitidine (range: 1 to 67) with a median follow up of 66.1 months. Median age was 76 years (range: 45 to 93). AML was secondary to a previous myeloid malignancy in 34% of the cases, MDS in 71 patients (25.4%) or myeloproliferative neoplasms in 24 patients (8.6%). AML was therapy-related in 46 patients (16.5%).

Cytogenetic risk was adverse in 135 patients (49.1%), including 54 patients with complete (-17) or partial (del17p) deletion of chromosome-17, chromosome containing the *TP53* locus (19.4%). *TP53* status was available before azacitidine treatment for 224 patients.

We detected a *TP53* mutation in 55 patients (24.6%) at a VAF threshold of 10% and 64 patients (28.6%, S1 Fig) at a threshold of 1%. The following analysis was done using a VAF threshold of 10%. *TP53* locus was deleted and/or mutated (*TP53* alteration) in 68 patients (30.4%). Patient characteristics according to *TP53* mutational status are summarized in Table 1. Compared to patients without *TP53* mutation, patients harboring a *TP53* mutation presented more often with altered performance status (ECOG score ≥2 in 26.4% *vs.* 42%, respectively, p = 0.037), had a lower baseline median platelet count (74 G/L *vs.* 46 G/L, respectively, p = 0.001) and higher rate of adverse cytogenetics (33.1% *vs.* 96.4%, respectively, p<0.001).

### Prognosis factors for overall survival under azacitidine

Among the 224 patients with available *TP53* status at baseline, we looked for factors affecting overall survival (Table 2). Older age (hazard ratio [HR] = 1.02; 95% CI = [1.00;1.04]; p = 0.040), a higher level of LDH (HR = 1.08; 95% CI = [1.04;1.11]; p<0.001), an adverse karyotype (HR = 1.79; 95% CI = [1.35;2.37]; p<0.001) and presence of *TP53* mutation (HR = 2.22; 95% CI = [1.60;3.08]; p<0.001) or alteration (*TP53* mutation and/or -17/17p-; HR = 2.53; 95% CI = [1.85–3.45]; p<0.001) were significantly associated with a poorer OS in univariable analysis (Table 2).

**Table 1. Patient characteristics according to TP53 status.**

| | Azacitidine cohort N = 279 | *TP53*wt N = 169 | *TP53*mut N = 55 | *TP53* unknown N = 55 | *TP53*wt vs *TP53*mut p value |
|---|---|---|---|---|---|
| **Baseline characteristics** | | | | | |
| **Median Age—years (range)** | 76 (45–93) | 76 (45–90) | 75 (50–86) | 76 (57–93) | 0.089 |
| **Male gender—n (%)** | 155 (55.6) | 100 (59.2) | 29 (52.7) | 26 (47.3) | 0.401 |
| **AML status—n (%)** | | | | | |
| *De novo* | 138 (49.5) | 92 (54.4) | 26 (47.3) | 20 (36.4) | |
| Secondary to MDS[a] | 71 (25.4) | 43 (25.4) | 9 (16.4) | 19 (34.5) | 0.084 |
| Secondary to MPN[b] | 24 (8.6) | 10 (5.9) | 7 (12.7) | 7 (12.7) | |
| Therapy related AML | 46 (16.5) | 24 (14.2) | 13 (23.6) | 9 (16.4) | |
| **ECOG performance status—n (%)** | | | | | |
| 0–1 | 168 (69.7) | 109 (73.6) | 29 (58.0) | 30 (69.8) | |
| 2–4 | 73 (30.3) | 39 (26.4) | 21(42.0) | 13 (30.2) | **0.037** |
| Unknown | 38 | 21 | 5 | 12 | |
| **Charlson score—n (%)** | | | | | |
| 0–1 | 176 (76.5) | 103 (73.0) | 37 (80.4) | 36 (83.7) | |
| >1 | 54 (23.5) | 38 (27.0) | 9 (19.6) | 7 (16.3) | 0.316 |
| Missing | 49 | 28 | 9 | 12 | |
| **Extramedullary disease-n (%)** | | | | | |
| No extramedullary disease | 228 (88.7) | 140 (88.6) | 46 (88.5) | 42 (89.4) | 0.977 |
| Extramedullary disease | 29 (11.3) | 18 (11.4) | 6 (11.5) | 5 (10.6) | |
| Missing | 22 | 11 | 3 | 8 | |
| **Median WBC[c] count (n = 274)—G/L (range)** | 2.7 (0.4–271.0) | 2.4 (0.7–122.7) | 2.3 (0.5–85.0) | 3.4 (0.4–271) | 0.440 |
| **Median platelet count (n = 274) -G/L (range)** | 67 (3–1271) | 74 (7–771) | 46 (3–1271) | 71.5 (5–736) | **0.001** |
| **Median LDH[d] (n = 260)—U/L (range)** | 540.5 (135–3525) | 502 (135–3525) | 569.5 (163–3175) | 662.5 (168–3503) | 0.149 |
| **Median % BM blast count (n = 272)–(range)** | 33 (0–85) | 35 (0–83) | 29.5 (9–85) | 28 (2–78) | 0.075 |
| **Albumin—n (%)** | | | | | |
| Normal | 164 (81.2) | 106 (84.8) | 32 (74.4) | 26 (76.5) | 0.125 |
| <Normal | 38 (18.8) | 19 (15.2) | 11 (25.6) | 8 (23.5) | |
| Missing | 77 | 44 | 12 | 21 | |
| **Cytogenetics (MRC[e])—n (%)** | | | | | |
| Non adverse | 140 (50.9) | 113 (66.9) | 2 (3.6) | 25 (49.0) | **<0.001** |
| Adverse | 135 (49.1) | 56 (33.1) | 53(96.4) | 26 (51.0) | |
| Unknown | 4 | 0 | 0 | 4 | |
| **Monosomal karyotype—n (%)** | 66 (24.6) | 14 (8.4) | 37 (67.3) | 15 (31.9) | **<0.001** |
| **Del17p or monosomy 17- n (%)** | 54 (19.4) | 13 (7.7) | 32 (58.2) | 9 (16.4) | **<0.001** |
| **Outcome** | | | | | |
| **Median number of azacitidine cycles n(range)** | 6 (1–67) | 8 (1–67) | 5 (1–22) | 6 (1–26) | **<0.001** |
| **Response** | | | | | 0.502 |
| CR[f]/CRi[g] –n(%) | 54 (19.4) | 30 (17.8) | 12 (21.8) | 12 (21.8) | |
| Failure–n(%) | 225 (80.6) | 139 (82.2) | 43 (78.2) | 43 (78.2) | |
| **Median duration of response–months [95%IC]** | 9.3 [6.7; 14.0] | 9.9 [6.7; 19.2] | 6.5 [4.4; 20.8] | 13.3 [1; NR] | 0.303 |
| **Median OS[h] --months [95%IC]** | 10.6 [9.7; 12.1] | 12.6 [10.3; 15.6] | 7.9 [3.1; 9.8] | 10.0 [5.1; 16.4] | **<0.001** |

[a]MDS myelodysplastic syndrome

[b]MPN myeloproliferative neoplasm

[c]WBC white blood cell

[d]LDH lactate deshydrogenase

[e]MRC Medical Research Council

[f]CR complete response

[g]RCi complete response with incomplete hematologic recovery

[h]OS overall survival.

**Table 2. Prognosis factors for overall survival in univariable analysis.**

|  | 6 mos.-OS (%) | HR [95%CI] | p-value |
|---|---|---|---|
| **Age** (continuous variable) |  | 1.02 [1.00; 1.04] | 0.040 |
| **Gender** |  |  |  |
| Male | 69 | 1.00 | 0.299 |
| Female | 71 | 0.86 [0.65; 1.14] |  |
| **AML status** |  |  |  |
| De novo | 74 | 1.00 | 0.558 |
| Secondary | 65 | 1.09 [0.82; 1.43] |  |
| **ECOG Performance status** |  |  |  |
| 0–1 | 72 | 1.00 | 0.071 |
| 2–4 | 60 | 1.35 [0.97; 1.86] |  |
| **Charlson score** |  |  |  |
| 0–1 | 70 | 1.00 | 0.852 |
| >1 | 72 | 0.97 [0.68; 1.38] |  |
| **Extramedullary disease** |  |  |  |
| No | 71 | 1.00 | 0.554 |
| Yes | 58 | 1.15 [0.73; 1.81] |  |
| **WBC count** (continuous variable) |  | 1.01 [1.00; 1.02] | 0.088 |
| **Platelets count** (continuous variable) |  | 0.87 [0.73; 1.04] | 0.117 |
| **LDH** (continuous variable) |  | 1.08 [1.04; 1.11] | <0.001 |
| **Albumin** |  |  |  |
| Normal | 72 | 1.00 | 0.099 |
| < Normal | 53 | 1.42 [0.93; 2.16] |  |
| **Cytogenetic risk (MRC)** |  |  |  |
| Non-adverse | 82 | 1.00 | <0.001 |
| Adverse | 57 | 1.79 [1.35; 2.37] |  |
| **TP53 mutation** |  |  |  |
| No | 75 | 1.00 | <0.001 |
| Yes | 53 | 2.22 [1.60; 3.08] |  |
| **TP53 alteration** |  |  |  |
| No | 79 | 1.00 | <0.001 |
| Yes | 49 | 2.53 [1.85; 3.45] |  |

In multivariable analysis: age (HR = 1.03; 95% CI = [1.01;1.05]; p = 0.001), LDH (HR = 1.07; 95% CI = [1.03–1.11]; p<0.001), adverse karyotype (HR = 1.58; 95% CI = [1.15–2.34]; p = 0.024) remained significantly associated with OS (Table 3).The effect of *TP53* mutation on OS was just below the threshold of statistical significance (HR = 1.49; 95% CI = [0.95–2.34]; p = 0.081).

## Patient outcome according to the *TP53* status

The 55 patients with *TP53* mutations had a significantly lower OS compared to wild-type *TP53* patients (Fig 1A; median OS: 7.9 months with *TP53* mutation *vs.* 12.6 months without; HR = 2.22; 95% CI = [1.60–3.08]; p<0.001). The 68 patients with *TP53* alteration (Fig 1B) had a worst OS compared to patients without (median OS: 5.4 vs. 14.0 months, respectively, HR = 2.53; 95% CI = [1.85–3.45]; p<0.001). Within the group of 109 patients with adverse karyotype, *TP53* mutation (Fig 1C; median OS 7.9 months *vs.* 9.6 months, respectively; HR = 1.61; 95% CI =

**Table 3. Prognosis factors for overall survival in multivariable analysis.**

|  | HR [95%CI] | *p-value* |
|---|---|---|
| **Age** (continuous variable) | 1.03 [1.01; 1.06] | *0.016* |
| **ECOG PS** | | |
| 0–1 | 1.00 | *0.838* |
| 2–4 | 0.96 [0.67; 1.39] | |
| **WBC count** (continuous variable) | 1.00 [0.99; 1.02] | *0.798* |
| **LDH** (continuous variable/100) | 1.07 [1.03; 1.11] | *<0.001* |
| **Platelets count** (continuous variable/100) | 0.94 [0.80; 1.09] | *0.395* |
| **Cytogenetic risk** | | |
| Non-adverse | 1.00 | *0.024* |
| Adverse | 1.58 [1.06; 2.34] | |
| ***TP53* mutation** | | |
| No | 1.00 | *0.081* |
| Yes | 1.49 [0.95; 2.34] | |

[1.08;2.41]; p = 0.019) and *TP53* alteration including mutation and/or deletion (Fig 1D; median OS 5.4 months *vs.* 11.2 months, respectively, HR = 2.03; 95% CI = [1.33–3.09]; p<0.001) remained significantly associated with poorer OS compared to patients with unaltered *TP53*.

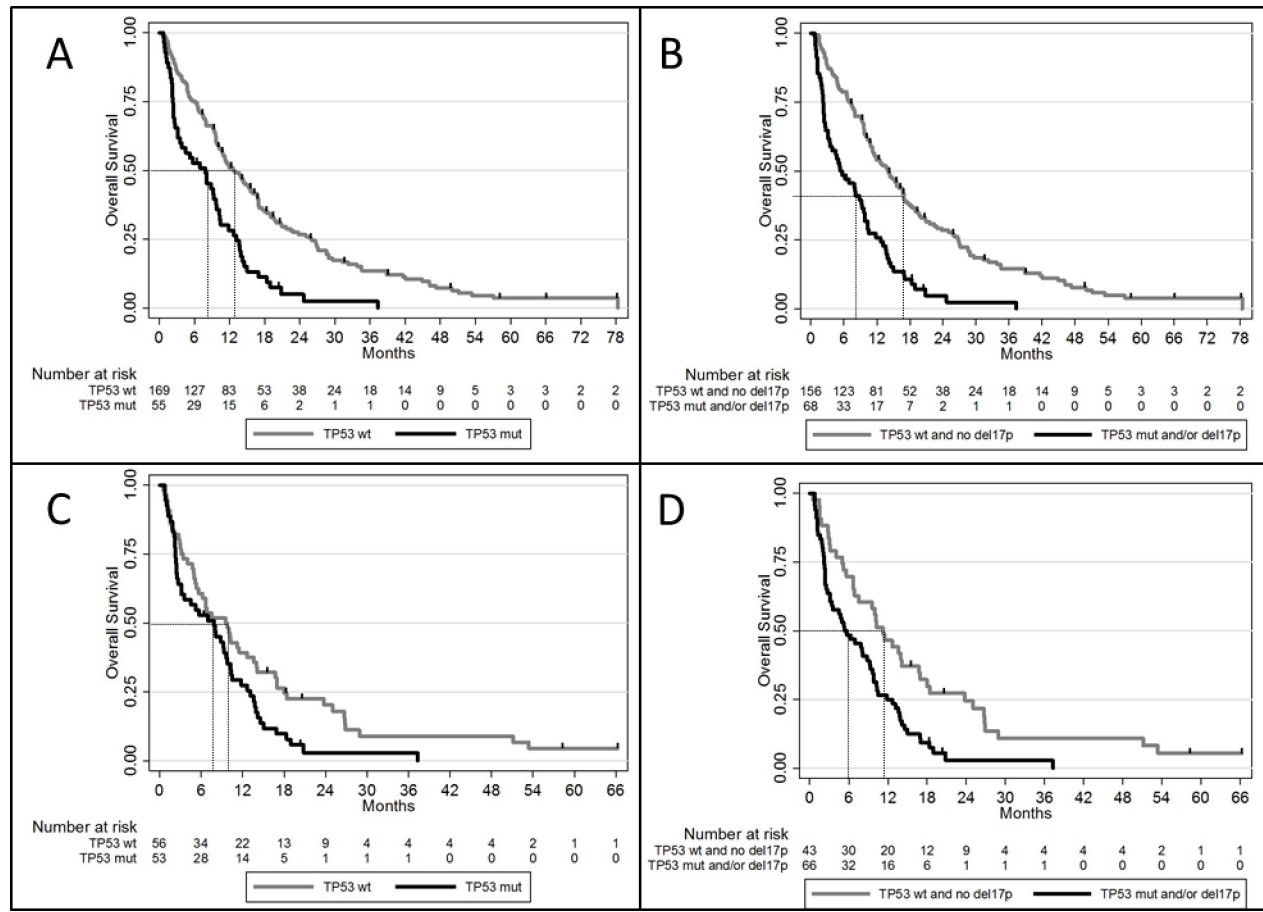

**Fig 1. Overall survival according to *TP53* alterations.** A. OS in patient with *TP53* mutation versus *TP53*^wt. B. OS in patient with *TP53* mutation and/or deletion versus patient without TP53 alteration. C. OS in patient with *TP53* mutation versus *TP53*^wt in the subgroup of adverse karyotype. D. OS in patient with *TP53* mutation and/or deletion versus patient without *TP53* alteration in the subgroup of adverse karyotype.

**Table 4. Univariable comparison of TP53 mutation functional characterization and response to azacitidine.**

| | Overall sample n = 55 | Responders N = 12 | Non responders N = 43 | p-value |
|---|---|---|---|---|
| **Disruptive *TP53* mutation** | | | | |
| Yes—n(%) | 15 (27.3) | 3 (25.0) | 12 (27.9) | 1.000 |
| No—n(%) | 40 (72.7) | 9 (75.0) | 31 (72.1) | |
| **Evolutionary Action score (n = 48),** | | | | |
| continuous variable 0–100 | 73.9 (28.1–95.7) | 79.3 (28.1–89.8) | 73.3 (48.5–95.7) | 1.000 |
| Median (range) | | | | |
| **Evolutionary Action score** | | | | |
| <75—n (%) | 25 (52.1) | 5 (45.5) | 20 (54.1) | 0.616 |
| ≥75—n (%) | 23 (47.9) | 6 (54.5) | 17 (45.9) | |
| Missing | 7 | 1 | 6 | |
| **Relative Fitness score (n = 54)**, log2 scale | | | | 0.675 |
| Median (range) | 0.094(-2.525–0.838) | 0.094(-0.789–0.579) | 0.094(-2.525–0.838) | |

In contrast to OS, response rates (CR/CRi) did not significantly differ according to the presence of *TP53* mutation (21.8% with *TP53* mutations *vs.* 17.8% without; p = 0.502) or alteration (19.1% with *vs.* 18.6% without; p = 0.926). Extending the definition of clinical response to partial responses (PR) or hematologic improvement (HI) did not modify the impact of *TP53* mutation in response to azacitidine (CR, CRi and PR: 23.6% with *vs.* 24.3% without; p = 0.925; CR, CRi, PR and HI: 36.4% with *vs.* 42.6% without; p = 0.414). Similarly, within the group of 109 patients with adverse karyotype, *TP53* mutation did not impact response achievement (20.8% RC/RCi with *TP53* mutation *vs.* 14.3% without; p = 0.374).

## Patient outcome according to TP53 mutation classifications

Among the 55 patients with a *TP53* mutation, we identified 49 cases (89%) with a unique *TP53* mutation (42 missenses [86%], 3 nonsenses [6%], and 4 frameshifts [8%]) and 6 cases (11%) with 2 mutations (2 patients with missense and frameshift mutations and 4 with 2 missense mutations).

As the impact of the *TP53* mutations is heterogeneous, we may assume that a specific subgroup of variants might be sensitive to HMA. We evaluated their impact on azacitidine response using three recent classifications of TP53 mutations [24, 25, 33]. Twelve of these patients were classified as responders and 43 as non-responders.

Disruptive mutations were detected in 15 patients (27.3%), classification based on the consequences of the *TP53* mutations relying on the location of the mutation and the predicted amino acid alterations. A TP53 Evolutionary Action Score was assessable for 48 patients (87%) and a relative flexible score derived from Kotler *et al.* for 54 patients (98.2%). Functional categorization of *TP53* variant is summarized in S2 Table. Comparison of these 3 different classifications of *TP53* mutations is summarized in Tables 4 and 5. None of these classifications were associated with response to azacitidine or OS (Fig 2).

## Discussion

Our study of 224 patients constitutes, to our knowledge, the largest cohort of elderly AML patients treated with HMA analyzed for *TP53* mutations so far. We identified an overall prevalence of 24.6%. of *TP53* mutations with a VAF >10%. Among them, 30% were localized in the main *TP53* hotspots for single base substitutions, which is in line with previous descriptions in solid tumors [30] and in AML [12]. The percentage of patients with *TP53* mutation in our cohort is higher than previously reported in elderly AML patients [36–38], which could be

**Table 5. Univariable comparison of *TP53* mutation functional characterization and overall survival.**

|  | Event/N | 6 mos.-OS (%) | HR [IC95%] | p-value |
|---|---|---|---|---|
| **Disruptive TP53 mutation** |  |  |  |  |
| No | 38/40 | 55 | 1.00 | 0.798 |
| Yes | 15/15 | 47 | 0.92 [0.50; 1.71] |  |
| **Evolutionary Action score (0–100)** |  |  |  |  |
| <75 | 24/25 | 48 | 1.00 | 0.923 |
| ≥75 | 22/23 | 61 | 0.97 [0.54; 1.75] |  |
| **Evolutionary Action score** (continuous variable, 0–100) |  |  | 1.01 [0.99; 1.03] | 0.515 |
| **Relative Fitness score** log2 scale (continuous) |  |  | 0.75 [0.45; 1.22] | 0.244 |

easily explained by the high proportion of adverse cytogenetics and secondary AML in this group of patients deemed unfit for IC.

The increasing knowledge of mutant forms of *TP53*, has provided detailed insights into the functional consequences of *TP53* mutations and supports the hypothesis that all *TP53* mutations are not functionally equivalent [23]. The majority of these mutations are missense mutations in the DBD and therefore lead to loss of target gene transactivation [39]. In addition to this loss of function, mutant TP53 may exhibit dominant negative effect on wild-type *TP53* [31] or gain-of-function properties with capacity of transactivating non-canonical target genes that confer selective growth advantage, migratory potential, and drug resistance [40]. Different approaches have been used to systematically categorize various mutant *TP53* forms, based on their functionality in tumor suppression. We selected 3 different classification systems [24, 25, 33] able to characterize *TP53* mutations and we compared this predicted phenotype with patient outcome under azacitidine. Although none of these predictive methods succeed in identifying *TP53* mutated AML patient who could benefit from azacitidine, it remains unknown whether this lack of reliability could be explained by the classification system, or by the biology of this subgroup of AML. It also remains unknown whether these classifications could distinguish *TP53* mutated AML patients with specific outcome treated with other therapy. With the potential differential effect of *TP53* status regarding decitabine or azacitidine therapy, it would be of great interest to investigate the accuracy of these phenotype-genotype tools in predicting the outcome of AML patients treated with decitabine. Given the difficulty of choice between intensive and low-intensity therapy one might also investigate the impact of these TP53 mutant classification system in patients treated with IC.

We did not find any association between *TP53* alterations (including mutation and/or deletion) and response to azacitidine but an association with shorter OS which was significant in univariable analysis (12.6 months in *TP53*wt *versus* 7.9 months in *TP53*mut [p<0.001]) and just below the threshold of significance in multivariable analysis (HR = 1.49; 95% CI = [0.95–2.34]; p = 0.081). This finding is comparable to recent data from phase II trial testing frontline decitabine in AML deemed unfit for IC [7, 41]. Survival outcomes in our cohort are also in line with the biomarker cohort of the phase 3 trial AZA-AML-001 [22], which have a median OS of 7.2 months in *TP53*mut patients compared to 12 months in *TP53*wt patients. This confirms that *TP53* mutations have limited impact on remission achievement in AML as in high-risk MDS but strongly affect OS [42, 43]. We could not reproduce results from Welch *et al.* [21] who reported *TP53* mutation as a positive prognosis factor for response to decitabine without survival advantage, raising the question whether decitabine should be preferred to azacitidine in *TP53* mutated AML patients. We assessed the impact of *TP53* mutation on response rate as defined by CR/CRi, while Welch *et al.* compared *TP53* mutational status and response defined by morphological leukemia free state (MLFS) rate after the first treatment cycle but it

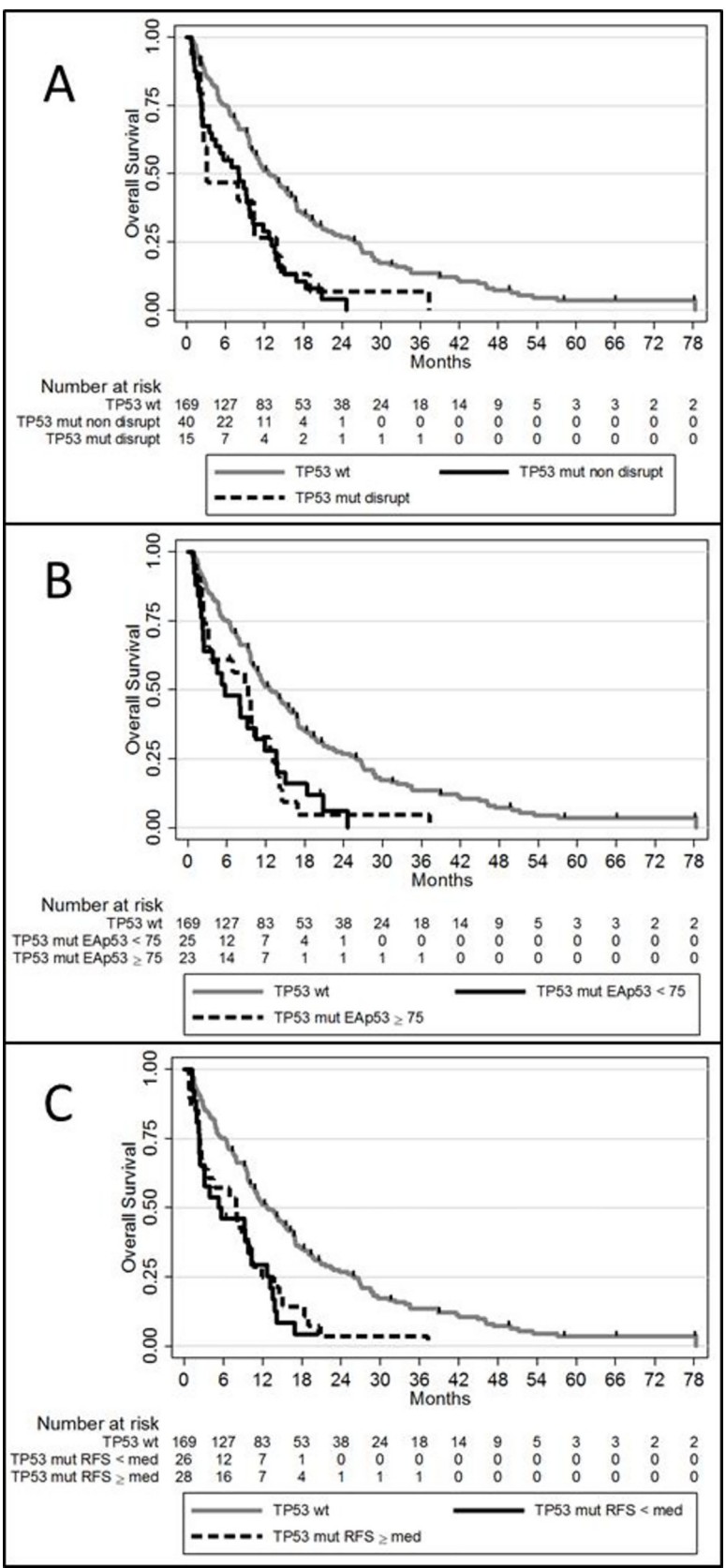

**Fig 2. Overall survival according to TP53 mutations classifications.** A. OS according to *TP53* disruptive classification: disruptive *TP53* mutation versus non-disruptive *TP53* mutation versus *TP53*^wt. B. OS according to evolutionary action score: EAp53 < 75 versus EAp53≥ 75 versus *TP53*^wt. C. OS according to relative flexible score: RFS < median versus ≥ median versus *TP53*^wt.

was not clear in their description of responders whether patient achieving MLFS after first cycle of decitabine eventually converted to a complete remission or improved OS. Of note, we did not find any impact of *TP53* mutation on overall response rate when response was defined with less stringent criteria including patients achieving at least HI. The differences could also rely on the doses of HMA administrated, Welch *et al.* used decitabine at a 200mg/m$^2$ divided daily dose for 10 days every 4 weeks, i.e. twice the dose of the current FDA approved scheme for high risk MDS and more than twice the monthly dose of the regimen tested by Lübbert *et al.* in elderly patients with AML [7]. Although, Blum *et al.* [44] reported improved CR rate in a phase II trial of elderly AML patients with this intensified scheme, the preferred dose and schedule of decitabine remains uncertain and is mainly limited by the myelotoxicity of the drug. Even though decitabine and azacitidine are both cytosine analogs with identical ring structure, they differ by the sugar attached to this ring. The deoxyribose in decitabine allows the incorporation of all metabolites to DNA, whereas only 10–20% of azacitidine is converted into a deoxyribonucleotide, the remaining of the drug being incorporated into RNA. The mechanism of action of both drugs is not fully understood and observed differences in outcome with decitabine compared to azacitidine for patient with specific genotype could presumably give information into precise mechanisms of action of these drugs.

In depth *TP53* genetic integrity analysis will also become inevitable for patients treated with FDA-approved association of HMA and BCL-2 inhibitor venetoclax. Recent data on molecular predictors of response with venetoclax combinations in older patients with AML indicate that TP53 loss promotes resistance to both venetoclax and chemotherapy with apparition of biallelic TP53 defectives clones at progression [45]. It remains unknown if a subset of TP53 abnormalities evase this selective pressure.

Regarding the growing field of *TP53*-activating compounds [46] and targeted therapy against *TP53* pathway genes [47] (*e.g.*, *MDM2*), a better characterization of mutational and non-mutational TP53 alterations will become useful in the initial workup of each AML patient [48]. In this regard, our cohort constitutes a reference for ongoing non-randomized phase II trial testing these *TP53*-activating compounds whose results are eagerly anticipated.

## Supporting information

**S1 Table. Primers used for TP53 targeted sequencing.**
(DOCX)

**S2 Table. *TP53* mutation functional characterization and patient outcome.**
(DOCX)

**S1 Fig. Overall survival according to *TP53* mutation with a threshold ≥1%.**
(DOCX)

## Acknowledgments

The authors would like to thank the data management unit of Toulouse University for its support enabling e-CRF. We thank all the members of the G.A.E.L (*Gaël Adolescent Espoir Leucémie*).

## Author Contributions

**Conceptualization:** Pierre Bories, Véronique De Mas, Christian Recher, Eric Delabesse.

**Data curation:** Pierre Bories, Naïs Prade, Stéphanie Lagarde, Julien Plenecassagnes, Isabelle Luquet, Véronique De Mas, Manon Cassou.

**Formal analysis:** Naïs Prade, Stéphanie Lagarde, Bastien Cabarrou, Laetitia Largeaud, Isabelle Luquet, Manon Cassou, Audrey Sarry, Célestine Simand.

**Investigation:** Pierre Bories.

**Methodology:** Pierre Bories, Bastien Cabarrou, Thomas Filleron, Eric Delabesse.

**Resources:** Audrey Sarry, Sarah Bertoli, Eric Delabesse.

**Software:** Naïs Prade, Stéphanie Lagarde, Julien Plenecassagnes.

**Supervision:** Eric Delabesse.

**Validation:** Laetitia Largeaud, Thomas Filleron, Luc-Matthieu Fornecker, Célestine Simand, Sarah Bertoli, Christian Recher, Eric Delabesse.

**Writing – original draft:** Pierre Bories.

**Writing – review & editing:** Pierre Bories, Bastien Cabarrou, Luc-Matthieu Fornecker, Sarah Bertoli, Christian Recher, Eric Delabesse.

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
