## [Decision Letter · Decision Letter 0]

9 Jun 2020

PONE-D-20-11937

Impact of TP53 mutations in acute myeloid leukemia patients treated with azacitidine

PLOS ONE

Dear Dr. Bories,

Thank you for submitting your manuscript to PLOS ONE. After careful consideration, we feel that it has merit but does not fully meet PLOS ONE’s publication criteria as it currently stands. Therefore, we invite you to submit a revised version of the manuscript that addresses the points raised during the review process by both Reviewers, experts in the AML field.

We look forward to receiving your revised manuscript.

Kind regards,

Francesco Bertolini, MD, PhD

Academic Editor

PLOS ONE

Journal Requirements:

2. In the Methods section of the manuscript, please indicate that Commission Nationale de l’Informatique et des Libertés (CNIL) authorised the use of patient data analyzed in your study."

3. In the ethics statement in the manuscript and in the online submission form, please provide additional information about the patient records used in your retrospective study, including: a) whether all data were fully anonymized before you accessed them; b) the date range (month and year) during which patients' medical records were accessed; and dc) the source of the medical records and tissue samples analyzed in this work (e.g. hospital, medical center or biobank name).

"This work was supported by the French government under the "Investissement d'avenir" program (ANR-11-PHUC-001)."

"The author(s) received no specific funding for this work"

"Christian Recher has received research funding from Celgene and served as

consultant for Celgene"

6. Please amend the manuscript submission data (via Edit Submission) to include author Stéphanie Lagarde.

7. Please amend your authorship list in your manuscript file to include author Stéphanie Dufréchou.

8. Please ensure that you refer to Figure 2 in your text as, if accepted, production will need this reference to link the reader to the figure.

Reviewers' comments:

Reviewer's Responses to Questions

**Comments to the Author**

1. Is the manuscript technically sound, and do the data support the conclusions?

Reviewer #1: Yes

Reviewer #2: Partly

2. Has the statistical analysis been performed appropriately and rigorously? 

Reviewer #1: No

Reviewer #2: No

3. Have the authors made all data underlying the findings in their manuscript fully available?

Reviewer #1: Yes

Reviewer #2: Yes

4. Is the manuscript presented in an intelligible fashion and written in standard English?

Reviewer #1: Yes

Reviewer #2: Yes

5. Review Comments to the Author

Reviewer #1: Bories et al submit a retrospective analysis of elderly AML patients treated with azacitidine who were considered unfit for induction chemotherapy. They identify 55 patients with TP53 mutations and 68 patients with mutations and/or deletions. The mutations are predominantly missense, organize around hotspots in the DNA-binding domain, are associated with near universal adverse risk karyotypes, all expected phenotypes for TP53 variants in AML. Of note, patients with TP53 mutations had worse performance status. Survival outcomes were worse with TP53 mutations or alterations. Overall, this is a well-written post-hoc analysis of a fairly large dataset. The results are interesting and the discussion is germane.

Concerns

Performance status > 2 is more common in TP53 mutated/altered cases. Poor performance status is commonly associated with poor survival in older AML studies. In the multivariate analysis, PS < 2 is used. It is unclear why the threshold is changed. The authors should repeat the multivariate analysis using > 2 as the cut off because this is associated with a phenotype among TP53 mutated patients and it would be important to know if poor performance status contributes to the adverse survival in the TP53 mutated cases.

Do the authors have access to the number of cycles of therapy received? Is this related to TP53 status, response, or survival?

Reviewer #2: This is an interesting retrospective study focusing on the impact of TP53 gene mutations in patients with AML treated with azacitidine in a quite large series of patients prospectively enrolled in the French regional healthcare network between 2007 and 2016 (n=279). The manuscript is well written and the data are clearly and straightforwardly presented, with well-designed tables and pictures. However, in its present version, I have a few major concern requiring some clarifications and possible refined analysis to be performed:

1. The main finding of this analysis is the association between the presence of any TP53 mutation and worse overall survival, with no impact on the treatment outcome. As it has been correctly stated and discussed also by the authors, this latter finding is in countertrend compared to the previously published results by Welch et al with decitabine. However, among the 224 patients who have been here retrospectively screened (by exome sequencing in 49 and NGS multiplex PCR in 179), TP53 mutations have been detected in 55 patients of whom 96.4% were also harbouring high-risk cytogenetics. In light of this strong correlation (adverse cytogenetics were found in only 33% of patients with TP53 WT), the inclusion of a possible interaction factor between the two variables in the multivariate model is highly recommended to verify their real independence. Should a significant interaction exist, it would be very important to verify its effect on the two variables taken individually.

2. Since the number of administered azacitidine cycles has been extremely heterogeneous (ranging from 1 to 67 with a median of 6), it would be also important to investigate the impact of the TP53 mutation on the treatment duration. In other words, what was the median number (and ranges) of administered cycles in the TP53wt, TP53mut and TP53unknown subgroups? Furthermore, were the azacitidine cycles all administered according to the 5-2-2 schedule, or there were also patients who received the 7 days in a raw schedule? If this is the case, any difference in the percentage of TP53 mutation among possible differently treated patients?

3. What was the median response duration among the twelve patients harboring a TP53 mutation and classified as responders? How does this median duration compare to that of responders in the TP53wt and TP53unknown groups?

Minor comments/questions:

1. Has been any post-treatment NGS analysis performed among responders to investigate on possible VAF variation for TP53mutations?

2. Tables 1 and 2 include “tumour” among enlisted variables. What does this mean? Patients with tumour are those with a previous/concomitant malignancy other than AML? Please, clarify.

6. PLOS authors have the option to publish the peer review history of their article (what does this mean?). If published, this will include your full peer review and any attached files.

Reviewer #1: No

Reviewer #2: Yes: Francesco Onida

---

## [Author Response · Author response to Decision Letter 0]

31 Jul 2020

We warmly thank the two reviewers for their comments. Whenever possible, the manuscript has been revised according to their comments and detailed explanations and answers are given in this cover letter. 

Additionally, an ancillary biomarker study of a phase II trial testing frontline decitabine in elderly AML patients have been recently published with similar results to ours (Becker et al, Annals of Hematology, June 2020). This was added in the discussion (line 279, reference 41).

Responses to reviewer 1:

1 Performance status > 2 is more common in TP53 mutated/altered cases. Poor performance status is commonly associated with poor survival in older AML studies. In the multivariate analysis, PS < 2 is used. It is unclear why the threshold is changed. The authors should repeat the multivariate analysis using > 2 as the cut off because this is associated with a phenotype among TP53 mutated patients and it would be important to know if poor performance status contributes to the adverse survival in the TP53 mutated cases.

We agree that AML patients with TP53 mutation commonly present with altered performance status (PS) which may be associated with shorter overall survival. In our cohort, among 55 TP53mut AML patients, 21 patients (42%) presented with baseline ECOG PS >1 but only 4 patients (7%) with baseline ECOG PS >2. In order not to unbalanced the groups, we decided to use the >1 cut-off. To avoid any ambiguity between the > and ≥ symbols we present results in table 1 and 2, with ECOG 0-1 and 2-4 groups.

As you suggested, we included ECOG PS in the multivariate analysis for OS which is relevant regarding its classical impact on OS and also justified by differences in term of baseline ECOG observed between TP53wt and TP53mut patients.

The inclusion of ECOG PS in the multivariable analysis for OS slightly modified the results of the model, notably with the association between TP53 mutation and shorter OS being just below the threshold of significance at 0.081, while the hazard ratio remained at 1.49. Accordingly, we modified the text in the result section (line195) and in the discussion (line 276)

2 Do the authors have access to the number of cycles of therapy received? Is this related to TP53 status, response, or survival?

We included the median number of azacitidine cycle received in table 1 for each cohort (global cohort, TP53wt, TP53mut, and TP53 unknown subgroups). The number of treatment cycle is significantly lower in the TP53mut group compared to the TP53wt group, which could be explained by a shorter duration of response in this group. Median duration of response has also been added in table 1, alongside with response rate and median overall survival in the outcome section.

Responses to reviewer 2:

1. The main finding of this analysis is the association between the presence of any TP53 mutation and worse overall survival, with no impact on the treatment outcome. As it has been correctly stated and discussed also by the authors, this latter finding is in countertrend compared to the previously published results by Welch et al with decitabine. However, among the 224 patients who have been here retrospectively screened (by exome sequencing in 49 and NGS multiplex PCR in 179), TP53 mutations have been detected in 55 patients of whom 96.4% were also harbouring high-risk cytogenetics. In light of this strong correlation (adverse cytogenetics were found in only 33% of patients with TP53 WT), the inclusion of a possible interaction factor between the two variables in the multivariate model is highly recommended to verify their real independence. Should a significant interaction exist, it would be very important to verify its effect on the two variables taken individually.

The required sample size to test an interaction with an acceptable statistical power is usually very large (at least four times more than when no interaction is assumed) (Brookes et al., Journal of Clinical Epidemiology 2004; Schmoor et al., Statistics in Medicine 2000; Peterson et al., Controlled Clinical Trials 1993). Given the size of our study and the number of patients in the subgroup with TP53 mutation and non-adverse cytogenetics (n=2), it was not appropriate to investigate the interaction between TP53 status and cytogenetics risk.

As an exploratory purpose, a subgroup analysis was then performed to assess the impact of TP53 status on overall survival in patients with adverse cytogenetics (median OS 7.9 months (mut) vs. 9.6 months (WT); HR=1.61; 95% CI=[1.08;2.41]; p=0.019, Fig 1C).

2) Since the number of administered azacitidine cycles has been extremely heterogeneous (ranging from 1 to 67 with a median of 6), it would be also important to investigate the impact of the TP53 mutation on the treatment duration. In other words, what was the median number (and ranges) of administered cycles in the TP53wt, TP53mut and TP53unknown subgroups? Furthermore, were the azacitidine cycles all administered according to the 5-2-2 schedule, or there were also patients who received the 7 days in a raw schedule? If this is the case, any difference in the percentage of TP53 mutation among possible differently treated patients?

We included the median number of azacitidine cycle received in table 1 in each cohort (global cohort, TP53wt, TP53mut, and TP53 unknown subgroups). The number of treatment cycle is significantly lower in the TP53mut group compared to the TP53wt group, which could be explained by a shorter duration of response in this group. Duration of response has also been added in table 1, alongside with response rate and median overall survival

In our daily practice azacitidine is mainly administered using the so-called 5-2-2 schedule. Some patients may also have received the 7 days in a raw schedule, in particular for the first cycle frequently performed as inpatient. Unfortunately, this information is not collected in the Toulouse AML database and we cannot assess the impact of azacitidine schedule on outcome of TP53wt versus TP53mut patients.

3) What was the median response duration among the twelve patients harboring a TP53 mutation and classified as responders? How does this median duration compare to that of responders in the TP53wt and TP53unknown groups?

Definition of duration of response was added in the method section (line 148). We add this information in table 1 (page 7). Median duration of response (DOR) was 9.3 months in the global cohort of responders (n=54), 9.9 months in patients with TP53wt (n=30), 6,5 months in patients with a TP53 mutation (n=12) and 13.3 months in patient with unknown TP53 status (n=12). Univariable comparison of the DOR between TP53wt and TP53mut patient was not statistically different (9.9 months versus 6.5 months, respectively, p=0.303), presumably due to the limited number of patients in each cohort.

Minor comments/questions:

1) Has been any post-treatment NGS analysis performed among responders to investigate on possible VAF variation for TP53mutations?

We acknowledge that TP53 mutation clearance would have add valuable information, unfortunately, such samples were not banked in daily practice.

2) 2. Tables 1 and 2 include “tumour” among enlisted variables. What does this mean? Patients with tumour are those with a previous/concomitant malignancy other than AML? Please, clarify 

In table 1 and 2 “Tumour” in fact designates clinical extra medullary disease such as splenomegaly, hepatomegaly, lymph nodes or gingival hypertrophy. We acknowledge this designation is ambiguous, and changed it in the manuscript and tables to “extramedullary disease” 

Journal requirements 

1. Our manuscript was entirely checked regarding PLOS ONE's style requirements and we made the following changes:

 - We used level 1 heading for all major sections (abstract, introduction…) with bold type 18pt, and sentence case, level 2 heading for sub-sections with 16pt bold type police, and level 3 heading with bold type 14pt police fur sub-section within section 2.

 -The term “Figure XX” was replaced by “Fig XX” in the whole manuscript and “Table S1” by “S1 Table”.

2. We add in the material section of the manuscript the following sentence: “The French Commission Nationale de l’Informatique et des Libertés (CNIL) authorised the use of patient data analyzed in our study” (line 95)

3. An ethics statement section was added at the end of the manuscript (line 478) and the Edit Submission. 

4. The sentence “This work was supported by the French government under the "Investissement d'avenir" program (ANR-11-PHUC-001).” (line 318) was removed from the acknowledgment section of the manuscript and this information was provided in the Funding Statement.

5. We add the sentence “This does not alter our adherence to PLOS ONE policies on sharing data and materials.” after "Christian Recher has received research funding from Celgene and served as consultant for Celgene" in the Competing Interests section form of the Edit Submission.

6. The manuscript was amended via Edit Submission to include author Stéphanie Lagarde (married name Stéphanie Dufrechou)

7. The manuscript file includes Stéphanie Lagarde as an author instead of Stéphanie Dufréchou

8. Reference to figure 2 was added in the manuscript file (line 241).

9. The section “supplementary material” was renamed “supporting information” and reformat according to PLOS One Supporting Information guidelines.

Addendum to the journal requirements on 07/31/ 2020: 

1. A minimal data set has been uploaded on Figshare repository. The following sentence has been added in the manuscript line 139 “All the data used for the analyses were deposited in Figshare: https://figshare.com/s/b86087d20fbd9634e156”. 

Separate captions of the supplementary files are included at the end of the manuscript

2. Please update the funding statement as follow: “This work was supported by the French government under the "Investissement d'avenir" program (ANR-11-PHUC-001). The funders had no role in study design, data collection and analysis, decision to publish, or preparation of the manuscript. During the period of the study Stéphanie Lagarde has received salary from the "Investissement d'avenir" program (ANR-11-PHUC-001). 

3/4. Copy of S1 table, S2 table and S1 Fig have been uploaded in the Edit submission.

---

## [Decision Letter · Decision Letter 1]

25 Aug 2020

Impact of TP53 mutations in acute myeloid leukemia patients treated with azacitidine

PONE-D-20-11937R1

Dear Dr. Bories,

We’re pleased to inform you that your manuscript has been judged scientifically suitable for publication and will be formally accepted for publication once it meets all outstanding technical requirements.

Kind regards,

Francesco Bertolini, MD, PhD

Academic Editor

PLOS ONE

Additional Editor Comments (optional):

Reviewers' comments:

Reviewer's Responses to Questions

**Comments to the Author**

1. If the authors have adequately addressed your comments raised in a previous round of review and you feel that this manuscript is now acceptable for publication, you may indicate that here to bypass the “Comments to the Author” section, enter your conflict of interest statement in the “Confidential to Editor” section, and submit your "Accept" recommendation.

Reviewer #2: All comments have been addressed

2. Is the manuscript technically sound, and do the data support the conclusions?

Reviewer #2: Yes

3. Has the statistical analysis been performed appropriately and rigorously? 

Reviewer #2: Yes

4. Have the authors made all data underlying the findings in their manuscript fully available?

Reviewer #2: Yes

5. Is the manuscript presented in an intelligible fashion and written in standard English?

Reviewer #2: Yes

6. Review Comments to the Author

Reviewer #2: All previous comments and suggestions have been adequately addressed in this revised version of the manuscript. I have no further comments.

7. PLOS authors have the option to publish the peer review history of their article (what does this mean?). If published, this will include your full peer review and any attached files.

Reviewer #2: **Yes: **Prof. Francesco Onida

---

## [Editor Report · Acceptance letter]

21 Sep 2020

PONE-D-20-11937R1 

Impact of *TP53* mutations in acute myeloid leukemia patients treated with azacitidine 

Dear Dr. Bories:

I'm pleased to inform you that your manuscript has been deemed suitable for publication in PLOS ONE. Congratulations! Your manuscript is now with our production department. 

Kind regards, 

on behalf of

Dr. Francesco Bertolini 

Academic Editor

PLOS ONE